

# Association of metallic elements with telomere length in children with autism spectrum disorder

Qiuyan Zhu[1,2], Tong Zhang[2], Yanan Sun[3], Jinming Liu[1], Zizi Liu[1], Fengxiang Wei[2] and Yu Jin[1]

[1] Department of Maternal and Child Health, School of Public Health, Sun Yat-Sen University, Guangzhou, China
[2] Longgang District Maternity & Child Healthcare Hospital of Shenzhen City (Longgang Maternity and Child Institute of Shantou University Medical College), Shenzhen, China
[3] Longgang Central Hospital of Shenzhen, Shenzhen, China

Corresponding authors
Yu Jin, jinyu@mail.sysu.edu.cn
Fengxiang Wei, haowei727499@163.com

## ABSTRACT

**Background:** Imbalances in metal elements have been identified as a potential risk factor for autism spectrum disorder (ASD), and shortened telomere length (TL) is commonly observed in children with ASD. Metal elements may influence telomere homeostasis through oxidative stress, which could contribute to the pathogenesis of autism. However, studies examining the combined effects of metal elements on TL in children with ASD are limited. To fill the gaps in the current literature, this study aimed to investigate the relationship between six metallic elements: manganese (Mn), copper (Cu), zinc (Zn), calcium (Ca), magnesium (Mg), and iron (Fe), and TL in the whole blood of children with ASD.
**Methods:** A total of 83 children with ASD and 95 typically developing children were recruited. TL was measured using digital PCR, while metal concentrations were assessed using inductively coupled plasma mass spectrometry (ICP-MS). Linear regression analysis was first conducted to explore the correlations between metal elements and TL in both groups. Additionally, Bayesian Kernel Machine Regression (BKMR) was used to further examine the combined effects and potential interactions of these metals on TL in the ASD group.
**Results:** In the ASD group, Ca was found to have a protective effect on TL ($\beta$ = 0.07, 95% CI [0.01–0.13], $P$ = 0.027). In contrast, Mg showed a protective effect on TL in the control group ($\beta$ = 0.10, 95% CI [0.01–0.18], $P$ = 0.027). The BKMR model revealed a significant positive combined effect of the metal mixtures on TL in the ASD group, with Ca having the largest individual effect (PIP = 0.45). Further analysis indicated that increases in Zn and Mn concentrations from the 25th to the 75th percentile were negatively correlated with TL, while higher concentrations of Cu, Ca, Mg, and Fe were positively associated with TL. No significant interactions among the metals were observed.
**Conclusions:** This study suggests a potential link between metallic elements and TL in children with ASD, with Ca having the greatest effect. Our findings highlight the potential benefits of appropriate calcium supplementation as a protective strategy for lengthening telomeres in children with ASD, emphasizing the importance of early nutritional interventions to improve their overall health.

# INTRODUCTION

Autism spectrum disorder (ASD) is a group of neurodevelopmental disorders characterized by impaired social communication, restricted interests, and repetitive stereotyped behaviors (*Williams & First, 2013*). Currently, the global prevalence rate of ASD is approximately 1% (*Zeidan et al., 2022*). As the prevalence of ASD continues to increase over time, it has imposed a significant burden on families and society. However, the etiology and pathogenesis remain unclear, highlighting the need for further in-depth research. Several studies have shown that ASD is not only genetically influenced but also closely associated with environmental factors, lifestyle, and other variables (*Ji, Liu & Yu, 2016*).

Recent studies have reported imbalances in the levels of metallic elements, including manganese (Mn), copper (Cu), zinc (Zn), calcium (Ca), magnesium (Mg), and iron (Fe), in children with ASD (*Chen & Guo, 2022*), along with a significant reduction in telomere length (TL) (*Zhang et al., 2023b*). The six metallic elements studied play crucial biological roles and are closely associated with neurodevelopmental disorders, including ASD. For instance, Mn is a key component of antioxidant enzymes (*Chen, Bornhorst & Aschner, 2018*). Both lower and higher levels of Mn disrupt early neurodevelopment (*Bhang et al., 2013*), while overexposure to Mn may cause neurotoxicity, impairing cognition and leading to behavioral issues (*Zoni, Albini & Lucchini, 2007*). Cu contributes to bone tissue formation and metabolism (*Grzeszczak, Kwiatkowski & Kosik-Bogacka, 2020*) and is vital for biological processes linked to ASD, such as immune function (*Kelley et al., 1995*) and placental development. Zn is essential for neuronal regulation, synaptic plasticity, learning, and memory (*Qi & Liu, 2019*), deficiency of zinc in mice models has been shown to disrupt neural tube closure (*Li, Zhang & Niswander, 2018*) and induce ASD-related behaviors, including reduced social interaction. Ca and Mg are fundamental for nerve conduction and muscle contraction (*Capozzi, Scambia & Lello, 2020*) and are involved in regulating glutamate-activated channels in neuronal membranes, which are strongly associated with ASD pathogenesis (*Saghazadeh et al., 2017*). Ca also enhances nerve impulse transmission, and low levels of Ca may cause irritability and anxiety, chronic deficiencies may contribute to psychiatric disorders in children (*Mahaffey, Gartside & Glueck, 1986*; *Fiłon, Ustymowicz-Farbiszewska & Krajewska-Kułak, 2020*). Lastly, Fe is an essential component of hemoglobin (*Grzeszczak, Kwiatkowski & Kosik-Bogacka, 2020*), is linked to cognitive and behavioral deficits when deficient (*McCann & Ames, 2007*), and correlates with the severity of emotional and behavioral problems in individuals with ASD (*Saghazadeh et al., 2017*). Telomeres are DNA-protein complexes located at the ends of chromosomes (*Chen, Redon & Lingner, 2012*; *Palm & de Lange, 2008*), primarily functioning to prevent intracellular DNA damage and maintain genomic integrity (*Darrow et al., 2016*; *Gavia-García et al., 2021*; *Revy, Kannengiesser & Bertuch, 2023*). Previous literature (*Fernandes, Dsouza & Khattar, 2021*) suggests that external environmental factors, including metallic

elements, polycyclic aromatic hydrocarbons, endocrine disruptors, and airborne particulate matter, may reduce TL by inducing oxidative stress and inflammation.

It is hypothesized that a potential association exists between metal levels and TL in children with ASD, and that exploring this relationship is essential for a deeper understanding of the pathogenesis of ASD. However, few studies have investigated the effects of exposure to metal mixtures on TL and evaluated the exposure-response relationship in children with ASD. Therefore, the present study aims to investigate the association between whole blood levels of six metal elements (Mn, Cu, Zn, Ca, Mg, Fe) and TL in children with ASD, to elucidate their potential relationship and offer a new perspective on the etiology of ASD.

# MATERIALS AND METHODS

## Study design and population

Eighty-three children diagnosed with ASD at the Maternal and Child Health Hospital of Longgang District, Shenzhen, between November 2021 and December 2023, and 95 typically developing children from the same period, were selected as case-control subjects for this study. The final diagnosis of ASD was made according to the criteria in the Diagnostic and Statistical Manual of Mental Disorders-5 (DSM-5), Fifth Edition, by two specialists with over three years of clinical experience, in conjunction with the developmental history of the children. Children with chromosomal or inherited metabolic disorders, other neurodevelopmental disorders, a history of psychotropic medication use, or those who had recently taken micronutrient supplements were excluded from the study population. The study was approved by the ethics committee of Maternal and Child Health Hospital of Longgang District, Shenzhen, China under the approval number LGFYYXLLL-2021-003. Written informed consent was obtained from the primary guardians of all participating children.

## Assessment of the child's basic condition and autism symptoms

The questionnaire collected data on socio-demographic information, basic details about the child, and parents' behavioral habits, which was completed by the child's caregiver. All children were screened using the Child Autism Rating Scale (CARS) and the Autism Behavior Checklist (ABC). The CARS scale (*Xiao et al., 2022*) assesses children with ASD in 15 items, including interpersonal relationships, imitation (of words and actions), emotional responses, and somatic application skills. Each item is rated on a scale from 1 to 4, with a total score of 30 or greater indicating a diagnosis of autism, less than 36 indicating mild to moderate autism, and a score of 36 or greater indicating severe autism. The ABC scale (*Wei et al., 2021*) consists of 57 items grouped into five sections: sensory, relating, body and object use, language, and social and self-help skills. Each item is scored on a scale from 1 to 4, with a total score of <53 considered negative for screening, a score between 53 and 67 considered positive for screening, and a score of ≥68 used to aid in the diagnosis of autism. A higher total score indicates more severe behavioral symptoms of autism.

## Detection of metal elements and TL

Inductively coupled plasma mass spectrometry (ICP-MS) was used to detect six metallic elements in children's whole blood: Mn, Cu, Zn, Ca, Mg and Fe. The whole blood specimen was placed on a blood mixer for thorough mixing, and 0.1 mL of blood was accurately measured into 1.9 mL of diluent (0.1% $HNO_3$ + 0.1% X-100), and mixed for 30 s before testing.

Genomic DNA extraction from peripheral blood mononuclear cells (PBMCs). The lymphocyte layer was isolated from 1 mL of peripheral blood using a lymphocyte isolation solution within 2 h after venous blood collection from children, conducted by a pediatric nurse using a 5-mL EDTA anticoagulation tube (Solebo, Beijing, China). The white membrane layer between the plasma and isolate was aspirated with a pasteurized pipette, and the lymphocytes were washed with PBS and resuspended to obtain the final lymphocyte suspension. Cellular DNA was extracted using magnetic bead separation on a Lab-Aid 820 Nucleic Acid Extractor (Xiamen Zhishen Biotechnology Co., Ltd., Xiamen, China). The quality and concentration of the extracted DNA samples were evaluated using a Qubit fluorometer, and the DNA was stored at −80 °C until further analysis. Employing Digital Polymerase Chain Reaction (dPCR) (Beijing Xinyi Biological Co., Ltd., Beijing, China) absolute quantitative TL. dPCR is a third-generation PCR technique developed from traditional PCR and real-time fluorescence quantitative PCR (qPCR). This method enables the direct quantification of TL in target sequences without requiring standard dilution curves, reaction efficiency calibrations, or reference samples, offering high accuracy and precision. Its distinct advantages in the absolute quantification of DNA molecules have led to widespread adoption in various precision medicine applications, with its validity supported by recent literature (*Quan, Sauzade & Brouzes, 2018*). To further ensure the accuracy of our experimental results, we incorporated an internal reference gene as a control. Primer designs were adapted from prior studies (*Gao et al., 2020*). The target gene was TL, and the internal reference gene was Hemoglobin Subunit Beta (HBB); the primer sequences were as follows: TL-Forward: 5′-CGGTTTG TTTGGGTTTGGGTTTGGGTTTGGGTTTGGGTT-3′, TL-Reverse: 5′-GGCTTGCCT TACCCTTACCCTTACCCTTACCCTTACCCT-3′, HBB-Forward: 5′-AGGAGAAGTCT GCCGTTACTG-3′, HBB-Reverse: 5′-CCGAGCACTTTCTTGCCATGA-3′. The reaction system was 30 μL, with the reaction conditions as follows: 95 °C for 3 min, 95 °C for 15 s, 55 °C for 20 s, 72 °C for 25 s, and 40 cycles.

## Statistical analyses

Epidata 3.1 software was used for data entry and database creation, and R version 4.3.3 (*R Core Team, 2024*) was applied for statistical analysis. Categorical variables were summarized by the number of cases and composition ratio, while continuous variables that conformed to a normal distribution were represented as the mean ± standard deviation ($\bar{x} \pm s$). Continuous variables not following a normal distribution were reported as M ($P_{25}$, $P_{75}$). Chi-square tests were conducted to compare the group differences on categorical variables, while independent samples t-test and the Mann-Whitney U test were conducted to compare continuous variables that underwent normality testing.

Linear regression analysis was employed to assess the effect of metal content on TL in both groups of children, using standardized TL as the dependent variable and the content of the six metallic elements as the independent variables. Based on previous literature and the results of univariate analyses, the following relevant covariates were included in our model, including children's age, height, weight, sex, outdoor exercise time, family income, balanced dietary status, and parental smoking status. Additionally, we investigated the dose-response relationships and combined exposure effects of six metal elements and TL levels in children with ASD using the Bayesian kernel machine regression (BKMR) model, implemented with the BKMR package in R. TL served as the outcome variable, the six metal elements were treated as a mixture of exposures, and the covariates described above were used as well. The model was run for 10,000 iterations. The dose-response relationship for combined exposures was plotted, and posterior selection probabilities and effect values for each exposure were estimated. The dose-response curves were plotted for each exposure, and a final curve was plotted for first-order interactions within the mixed exposure. The variable q.fixed represents a specific percentile of other mixed exposures fixed during exposure calculations, while est provides the overall risk estimate of the mixed exposures. Posterior inclusion probability (PIP) indicates the relative importance of each exposure, with values closer to one representing higher importance. Significance was set at $P < 0.05$.

## RESULTS

### Sample characteristics

No statistically significant difference was observed in the mean age at entry between the two groups ($t = 0.67$, $P = 0.502$). The mean TL were 5,109.11 ± 2,028.10 kb and 6,230.1 ± 2,931.73 kb for each group, respectively, with a statistically significant difference between groups ($t = 3.00$, $P = 0.003$). No statistically significant differences were observed in height, weight, gender, ethnicity, birth weight, parental occupation, or daily outdoor playtime between groups ($P > 0.05$), However, statistically significant differences were identified in levels of Mn, Cu, Ca and Mg ($P < 0.05$), while no significant differences were found for Zn or Fe ($P > 0.05$) (Table 1). Compared to the typically developing group, the ASD group exhibited shorter TL, higher concentrations of Mn and Mg, and lower concentrations of Cu and Ca.

The CARS scale and ABC scale scores are shown in Table 2. The mean CARS scale score of 32.95 ± 4.76 suggests that children with ASD exhibit mild to moderate symptom severity. The mean ABC scale score of 54.59 ± 15.07 meets the positive diagnostic criterion, confirming the presence of ASD-related symptoms. The sensory score of 7.81 ± 4.54 indicates that children with ASD experience some mild sensory processing difficulties. The high relating score of 16.09 ± 5.76 reflects significant social interaction challenges, a core feature of ASD. Body and object use scores (6.91 ± 6.43) were relatively low but showed variability, suggesting that while some children with ASD may experience mild motor skill difficulties, not all have significant issues. The language score of 14.09 ± 5.68 was high, highlighting notable language development challenges, a common characteristic of autism. Social and self-help skills scores of 9.95 ± 4.62 were moderate, suggesting variability in

**Table 1 Basic demographics of children in the two groups.**

| | ASD (*n* = 83) | TD (*n* = 95) | $t/X^2/z$ | *P* |
|---|---|---|---|---|
| Age | 4.98 ± 2.10 | 4.81 ± 0.83 | 0.67 | 0.502 |
| Sex, *n* (%) | | | 0.01 | 0.958 |
| Male | 71 (0.85) | 81 (0.85) | | |
| Female | 12 (0.15) | 14 (0.15) | | |
| Height (cm) | 108.65 ± 13.53 | 108.64 ± 5.91 | 0.01 | 0.990 |
| Weight (kg) | 20.56 ± 8.27 | 20.14 ± 6.22 | 0.38 | 0.708 |
| Ethnicity, *n* (%) | | | 0.04 | 1.000 |
| Han | 79 (0.95) | 91 (0.96) | | |
| Other | 4 (0.05) | 4 (0.04) | | |
| Birth weight (kg) | 3.17 ± 0.50 | 3.26 ± 0.41 | 1.33 | 0.187 |
| Telomere length (Kb) | 5,109.11 ± 2,028.10 | 6,230.10 ± 2,931.73 | 3.00 | 0.003 |
| Mn (ug/L) | 12.80 (10.70, 14.90) | 11.60 (9.90, 13.20) | 2.99 | 0.003 |
| Cu (ug/L) | 986.80 (915.90, 1,065) | 1,028.2 (969.00, 1,123.90) | 2.25 | 0.024 |
| Zn (mg/L) | 4.70 (4.30, 5.00) | 4.80 (4.40, 5.20) | 1.03 | 0.299 |
| Ca (mg/L) | 64.90 (62.40, 67.30) | 66.20 (63.90, 69.20) | 2.49 | 0.013 |
| Mg (mg/L) | 39.90 (37.90, 41.20) | 37.70 (35.90, 39.50) | 4.20 | <0.001 |
| Fe (mg/L) | 448.30 (426.60, 470.00) | 447.10 (427.00, 470.00) | 0.23 | 0.819 |
| Daily playtime outside the home (h) | 1.70 ± 0.88 | 1.76 ± 0.88 | 0.43 | 0.667 |
| Father smoking, *n* (%) | | | 0.92 | 0.337 |
| Yes | 31 (0.37) | 29 (0.31) | | |
| No | 52 (0.63) | 66 (0.69) | | |
| Mother smoking, *n* (%) | | | | |
| Yes | 0 (0.00) | 0 (0.00) | | |
| No | 83 (1.00) | 95 (1.00) | | |
| Father's occupation, *n* (%) | | | 1.06 | 0.304 |
| Blue collar | 12 (0.14) | 9 (0.09) | | |
| White collar | 71 (0.86) | 86 (0.91) | | |
| Mother's occupation, *n* (%) | | | 0.38 | 0.540 |
| Blue collar | 5 (0.06) | 8 (0.08) | | |
| White collar | 78 (0.94) | 87 (0.92) | | |
| Father's education, *n* (%) | | | 0.64 | 0.422 |
| High school and below | 68 (0.72) | 82 (0.86) | | |
| Undergraduate and above | 15 (0.28) | 13 (0.14) | | |
| Mother's education, *n* (%) | | | 11.46 | 0.001 |
| High school and below | 60 (0.72) | 87 (0.92) | | |
| Undergraduate and above | 23 (0.28) | 8 (0.08) | | |
| Mothers' living conditions during pregnancy, *n* (%) | | | 14.30 | <0.001 |
| Quiet | 59 (0.71) | 88 (0.93) | | |
| Noisy | 24 (0.29) | 7 (0.07) | | |
| Annual family income, *n* (%) | | | 32.07 | <0.001 |
| <300,000 | 64 (0.77) | 33 (0.35) | | |

|  | ASD (*n* = 83) | TD (*n* = 95) | $t/X^2/z$ | *P* |
|---|---|---|---|---|
| ≥300,000 | 19 (0.23) | 62 (0.65) | | |
| Daily screen time, *n* (%) | | | 6.73 | 0.010 |
| ≤2 h | 70 (0.84) | 91 (0.96) | | |
| >2 h | 13 (0.16) | 4 (0.04) | | |
| Balanced diet, *n* (%) | | | 4.36 | 0.037 |
| Yes | 57 (0.69) | 78 (0.82) | | |
| No | 26 (0.31) | 17 (0.18) | | |

**Note:**
Continuous variables were presented as mean ± SD. Categorical variables were presented as *n* (%). SD, Standard deviation; *n*, numbers of subjects; %, percentage.

**Table 2  ABC and CARS scale scores for children with ASD.**

| Variables | (mean ± SD) |
|---|---|
| CARS score | 32.95 ± 4.76 |
| ABC score | 54.59 ± 15.07 |
| Sensory | 7.81 ± 4.54 |
| Relating | 16.09 ± 5.76 |
| Body and object use | 6.91 ± 6.43 |
| Language | 14.09 ± 5.68 |
| Social and self-help skills | 9.95 ± 4.62 |

**Note:**
SD, Standard deviation, ABC scale included five domains: sensory, relating, body and object use, language, and social and self-help skills.

self-care abilities, with some children encountering general challenges. Overall, children with ASD exhibited mild to moderate symptom severity, with pronounced difficulties in social interaction and language development. Additionally, difficulty levels varied across sensory processing, body and object use, social and self-help skills.

## Correlation between metallic elements and TL in two groups of children

Ca showed a protective effect against TL in the ASD group ($\beta$ = 0.07, 95% CI [0.01–0.13], *P* = 0.027), while Mg had a similar effect in the control group ($\beta$ = 0.10, 95% CI [0.01–0.18], *P* = 0.027). The variance inflation factor (VIF) indicated no covariance among the six metallic elements in the model (Table 3).

## Combined effect of metal exposure on TL in the BKMR model in ASD group

The BKMR model was employed to quantify the combined influence of six metallic elements on TL in individuals with ASD. The conditional posterior inclusion probabilities (condPIP) for each metal, as a measure of their involvement, are shown in Table 4. The findings suggest that Ca may play a pivotal role in the composite effects of multiple metal exposures on TL. Notably, the model revealed a statistically significant positive effect of the

**Table 3 Association of metal elements with TL in two groups of children.**

|  | ASD group | | TD group | |
|---|---|---|---|---|
|  | β (95% CI) | P | β (95% CI) | P |
| Mn | 0.01 [−0.06–0.06] | 0.988 | −0.05 [−0.14–0.04] | 0.279 |
| Cu | 0.01 [−0.01–0.02] | 0.195 | −0.01 [−0.01–0.02] | 0.260 |
| Zn | −0.22 [−0.51–0.07] | 0.129 | 0.27 [−0.01–0.56] | 0.060 |
| Ca | 0.07 [0.01–0.13] | 0.027* | 0.06 [−0.03–0.15] | 0.203 |
| Mg | 0.02 [−0.06–0.09] | 0.691 | 0.10 [0.01–0.18] | 0.027* |
| Fe | 0.01 [−0.01–0.01] | 0.507 | 0.01 [−0.01–0.02] | 0.717 |

Notes:
\* $P < 0.05$.
ASD, Autism spectrum disorder; TD, typically developing.

**Table 4 Posterior inclusion probability (PIP) for conditional inclusion from BKMR model in Autism.**

| Metals | PIP |
|---|---|
| Mn | 0.25 |
| Cu | 0.22 |
| Zn | 0.29 |
| Ca | 0.45 |
| Mg | 0.24 |
| Fe | 0.19 |

Note:
PIP, Posterior inclusion probability.

metal mixture on TL. Compared to the median (50th percentile) exposure level, significant combined effects were observed at alternative percentiles, as depicted in Fig. 1A.

When concentrations of other metals were held constant at their 25th, 50th, and 75th percentile values, a pronounced negative correlation emerged between changes in Zn and Mn concentrations (spanning the 25th to 75th percentiles) and TL (Fig. 1B). Conversely, a significant positive correlation was noted between changes in Cu, Ca, Mg, and Fe concentrations across the same percentile range and TL (Fig. 1B). To assess potential nonlinearity in the exposure-response relationship for each metal in isolation, univariate analyses were conducted while holding other exposures at their median levels. The results revealed positive exposure-response associations between TL and the levels of Cu, Ca, Mg, and Fe (Fig. 1C). In contrast, inverse exposure-response relationships were found between TL and Mn and Zn levels (Fig. 1C). No significant interactions among metals in their effects on TL were observed (Fig. 1D).

## DISCUSSION

The current results indicated a significant effect of Ca on TL in the ASD group, while Mg showed a more pronounced effect on TL in the control group. The BKMR model revealed that the metal mixture has a significant positive combined effect on TL in children with ASD, with Ca contributing most to this combined effect. There was a significant negative

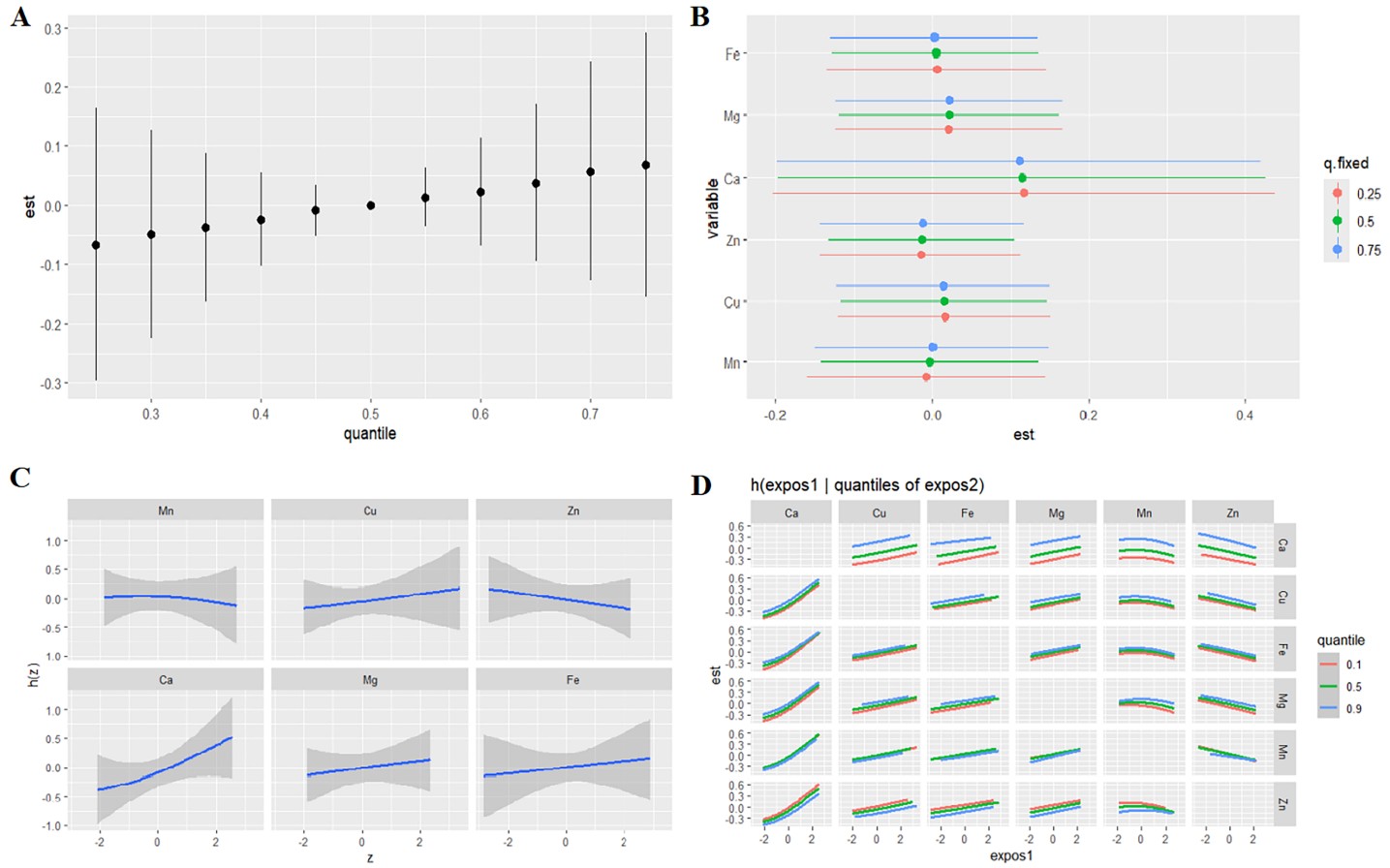

**Figure 1 Relationship between metals and TL in children with ASD analyzed by BKMR model.** (A) Combined effects of the metal as a mixture on TL in autism. (B) The single-exposure effect (estimates and 95% credible intervals) when all the other metals were fixed at 25th, 50th, and 75th percentiles. (C) Univariate exposure-response functions and 95% confidence interval for each metal with the other metals fixed at the median in autism. (D) Estimated effects (95% CIs) of single metal on TL in autism when the levels of other metals were fixing at 25th, 50th, and 75th. Models adjusted for child's age, gender, height, weight, daily playtime outside the home, balanced diet, annual family income, parental smoking status.

association between Levels of Zn and Mn and TL, whereas Cu, Ca, Mg, and Fe showed positive associations with TL.

Our findings align with previous studies (*Ma et al., 2022*) indicating that Mg levels are significantly higher in children with ASD than in typically developing children. However, other reviews (*Saghazadeh et al., 2017*) showed inconsistencies by observing severe Mg deficiencies in individuals with ASD. Furthermore, we found elevated levels of Mn, Mg, and Fe in children with ASD, whereas Cu, Zn, and Ca levels were lower in typically developing children. While there were inconsistent findings on metal levels in children with ASD, the literature indicates that deficiencies in Cu, Zn, Ca, Mg, and Fe are associated with ASD (*Behl, Mehta & Pandey, 2020*). These differences may be attributed to variations in sample size, population diversity, and geographic location, as well as nutritional, economic, and physical health factors. Thus, continued monitoring of metal levels in children with ASD is necessary. Additionally, our results showed shortened TL in children with ASD, consistent with previous findings (*Lewis et al., 2020*) indicating a correlation

between TL and cognitive and sensory functions. Telomere maintenance is crucial for stem cell function in the brain, with telomere shortening recognized as a potential factor in the pathogenesis of several neurological disorders, including ASD and schizophrenia, highlighting the importance of telomere homeostasis for children's health.

Metal elements perform essential biological functions in the human body, including maintaining central nervous system structure and function, supporting immune system activity, and promoting growth and development. In recent years, increasing attention has been directed toward the association between metal exposure and the risk of developing ASD; however, further studies exploring the association between metal exposure and TL are required. Occupational lead exposure has been associated with shortened TL in workers (Zhang et al., 2020), suggesting that high lead exposure may impair TL. In diabetic patients, a non-linear association between Cu and Zn and TL was observed (Zhang et al., 2023a), while Mg supplementation positively impacted TL maintenance and lifespan (Rowe, 2012). Steele et al. (2021) found that Fe supplementation had no significant impact on TL in subjects.

Our study indicated a significant effect of Ca on TL in children with ASD, consistent with findings from prior research (Lv, 2023), that mixed metal exposure was associated with increased TL in children, with selenium as the primary contributor and showing a linear exposure-response relationship with TL. Based on vitamin D's (VD) antioxidant properties and its close relationship with Ca ions in bone development and calcium-dependent signaling (Gusso et al., 2023), we hypothesize that Ca may function similarly to VD, offering a protective effect on TL in children with ASD. As cells divide, telomeres progressively shorten, thus driving cellular and individual senescence (Chen et al., 2024). Ca and Ca signaling have been recognized to be critical in regulating cellular aging processes over the past decade (Martin et al., 2023). Additionally, Ca signaling is essential for normal central nervous system functioning and is implicated in the pathophysiology of ASD (Pourtavakoli & Ghafouri-Fard, 2022). Collectively, this evidence further supports the conclusions of the present study.

Given the complex pattern of mixed metal exposure, we applied a novel statistical approach, the BKMR model, to evaluate the cumulative effects of metal levels on TL in children with ASD. To the best of our knowledge, this is the first study to apply this approach to evaluate the effects of combined metal exposure on TL in children with ASD. Notably, our results indicate that metal mixtures exert a significant positive effect on TL in children with ASD, with Ca as a primary contributor to this association. In the BKMR model, Zn and Mn levels were found to have a negative association with TL in children with ASD, while Cu, Ca, Mg, and Fe exhibited a positive association. Moreover, the BKMR model revealed that the effects of Ca and Fe on TL in children with ASD diminished as the concentrations of other metals approached the 75th percentile. Therefore, the relationship between Ca and Fe levels and TL in children with ASD appeared to be modulated by other metals, with stronger effects observed at lower concentrations of co-exposed metals. Nonetheless, future studies are needed to substantiate our findings. In summary, our findings underscore the importance of assessing cumulative metal exposures.

This study has several strengths. Epidemiological research focusing on the relationship between metal exposure and TL in children with ASD is limited. To the best of our knowledge, this is the first study to examine the correlation between Mn, Cu, Zn, Ca, Mg, and Fe levels and TL in children with ASD, potentially offering new insights into ASD pathogenesis. Furthermore, this study applied a novel mixture modeling approach, the BKMR model, to analyze the association between metal exposure and TL in children with ASD, enhancing the reliability of results for estimating the health effects of environmental pollutants. Finally, the study suggests that Ca may exert a protective effect on TL in children with ASD, presenting new research avenues and suggesting that appropriate Ca supplementation could help preserve TL and support healthy development in these children.

However, the present study has some limitations. Firstly, the case-control design is limited in establishing a causal relationship between metallic elements and TL in ASD offspring. Additionally, due to resource and technical constraints, only six common metallic elements were included in this study. Moreover, other factors could potentially influence telomeres, and confounders were assessed based on a questionnaire completed by the primary caregiver, which might also lead to recall bias. Consequently, further research is needed to explore the effects of metal element levels on TL in children with ASD. Future studies should aim to validate and investigate specific mechanisms, ideally with larger sample sizes and a broader range of metal elements.

## CONCLUSIONS

To our knowledge, this is the first study to assess the effects of mixed metal exposure on TL in children with ASD using a novel, robust BKMR model. The results indicated that the six metal groups had a significant positive combined effect on TL, with Ca contributing prominently to this overall effect. Zn and Mn concentrations were negatively associated with TL, while Cu, Ca, Mg, and Fe levels were positively associated. Given the ubiquity of metals and the association between TL, life expectancy, and chronic disease risk in children, these findings may have significant public health implications, highlighting the necessity of early nutritional interventions in enhancing children's health and quality of life.

## ACKNOWLEDGEMENTS

We would like to thank all the clinical staff at the Maternity and Child Healthcare Hospital of Longgang District, Shenzhen, who provided cooperation and assistance during the survey process, as well as all the parents and children who participated in the project.

### Funding

This work was supported by the following grants: Guangdong Provincial Fundamental and Applied Basic Research Fund Project (2023A1515012442), Special Project for Prevention and Rehabilitation of Hearing and Speech Disabilities of China Disabled Persons' Federation (2022CDPFHS-13), Key Project of the National People's Congress of Longgang

District, Shenzhen (LGCG2021164576), Shenzhen Science and Technology Program Project ( JCYJ20220530162412029), Shenzhen Science and Technology Program Project of Shenzhen Science and Technology Commission (KJYY201807703173402020), National Key Research and Development Program of China Special Research Project on Reproductive Health and Prevention and Control of Major Birth Defects (2016YFC1000103). The funders had no role in study design, data collection and analysis, decision to publish, or preparation of the manuscript.

## Grant Disclosures

The following grant information was disclosed by the authors:
Guangdong Provincial Fundamental and Applied Basic Research Fund Project: 2023A1515012442.
Special Project for Prevention and Rehabilitation of Hearing and Speech Disabilities of China Disabled Persons' Federation: 2022CDPFHS-13.
Key Project of the National People's Congress of Longgang District, Shenzhen: LGCG2021164576.
Shenzhen Science and Technology Program Project: JCYJ20220530162412029.
Shenzhen Science and Technology Program Project of Shenzhen Science and Technology Commission: KJYY20180703173402020.
National Key Research and Development Program of China Special Research Project on Reproductive Health and Prevention and Control of Major Birth Defects: 2016YFC1000103.

## Competing Interests

The authors declare that they have no competing interests.

## Author Contributions

- Qiuyan Zhu conceived and designed the experiments, performed the experiments, analyzed the data, prepared figures and/or tables, authored or reviewed drafts of the article, and approved the final draft.
- Tong Zhang conceived and designed the experiments, performed the experiments, prepared figures and/or tables, and approved the final draft.
- Yanan Sun conceived and designed the experiments, performed the experiments, prepared figures and/or tables, and approved the final draft.
- Jinming Liu analyzed the data, prepared figures and/or tables, and approved the final draft.
- Zizi Liu analyzed the data, prepared figures and/or tables, and approved the final draft.
- Fengxiang Wei conceived and designed the experiments, authored or reviewed drafts of the article, and approved the final draft.
- Yu Jin conceived and designed the experiments, authored or reviewed drafts of the article, and approved the final draft.

## Human Ethics

The following information was supplied relating to ethical approvals (*i.e.*, approving body and any reference numbers):

The study was approved by the Maternal and Child Health Hospital of Longgang District, Shenzhen, China under the approval number LGFYYXLLL-2021-003.

## Data Availability

The raw data is available in the Supplemental Files.

## Supplemental Information

Supplemental information for this article can be found online at http://dx.doi.org/10.7717/peerj.19174#supplemental-information.

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
