# Peer review of "Association of metallic elements with telomere length in children with autism spectrum disorder"

_PeerJ, doi:10.7717/peerj.19174_

## Round 0.1 · original submission · Major Revisions

This submission was previously given an 'open rejection' decision and the authors have resubmitted.

After the revisions, the manuscript has been significantly improved, however, some additional edits are needed before it could be considered for publication.

·

Basic reporting

I am satisfied with the answers provided by authors.

Experimental design

Authors explained restrictions concerning experimental design.
I accept their explanation.

Validity of the findings

Revised text is good and I accept authors explanation.

Additional comments

No additional comments

Reviewer 2 ·

Basic reporting

This is a revision of a previous submission. In this revised manuscript, authors assess the effects of six metal elements on TL in children with ASD, identifying Ca as a vital positive contributor to TL within the metal mixture using BKMR. After revision, this manuscript has been dramatically improved; however, a few edits are needed before it could be considered for publication.

Experimental design

See Additional comments

Validity of the findings

See Additional comments

Additional comments

According to the methods, TL is only quantitated in the lymphocytes rather than the whole WBC. Please justify in the Introduction why only the adaptive arm is investigated, especially separating WBC is much easier than separating lymphocytes from whole blood.

L114: “Detection of metal elements and TL” -- it matches the order below.

L115: “six metallic”

It is missing how the obtained dPCR data is analyzed. How the TL is evaluated using the dPCR results if a standard curve is not acquired. Are the TL primers correct???

L119: The first comma is suspected to be a full stop. Rewrite the sentence before that comma.

L129-130: “forward primer” and “reverse primer”. Also, the four “primer” are not necessary.

Spell HBB

L 151, “and the covariates described above were used as well”. “Previously described” usually refers to a published work.

In the Sample characteristics section, the no sig parameters (age, height, weight, gender ect.) could be put together aiming to show the comparability of the samples. There are sig on diet, mother’s education, and mother’s living conditions during pregnancy. Why are some not selected as covariates during statistical analysis? And how would the sig parameters potentially affect TL?

What are the bottom 5 lines in Table 2? Are they subsets of either CARS or ABC scores? If so, why are only these 5 otherwise listed out of 15/57??? If they are elevated other than CARS or ABC, give a full description in the Methods section.

The positive correlation between Mg and TL is subtle (Fig 1B). Please make this panel clear.

L293: “clinical staff”

---

## Round 0.2 · accepted · Accept

In the revised version the authors took into consideration all comments and remarks. I recommend to accept the manuscript for publication in PeerJ.

Reviewer 2 ·

Basic reporting

The is the second revision of a previous manuscript. The authors revised the manuscript carefully and thoroughly based upon reviewers' comments. This version is about ready for publication

Experimental design

No comment

Validity of the findings

No comment